# Development of Type 1 Diabetes in Mice Is Associated with a Decrease in IL-2-Producing ILC3 and FoxP3^+^ Treg in the Small Intestine

**DOI:** 10.3390/molecules28083366

**Published:** 2023-04-11

**Authors:** Tamara Saksida, Verica Paunović, Ivan Koprivica, Dragica Mićanović, Bojan Jevtić, Natalija Jonić, Ivana Stojanović, Nada Pejnović

**Affiliations:** 1Department of Immunology, Institute for Biological Research “Siniša Stanković”—National Institute of Republic of Serbia, University of Belgrade, Bulevar Despota Stefana 142, 11060 Belgrade, Serbia; cvjetica@ibiss.bg.ac.rs (T.S.); ivan.koprivica@yahoo.com (I.K.); gajic_dragica@yahoo.com (D.M.); bojanbh@gmail.com (B.J.); jonic.natalija@gmail.com (N.J.); nada.pejnovic@gmail.com (N.P.); 2Institute of Microbiology and Immunology, Faculty of Medicine, University of Belgrade, Pasterova 2, 11000 Belgrade, Serbia; vericapaunovic@gmail.com

**Keywords:** type 1 diabetes, type 3 innate lymphoid cells, regulatory T cells, interleukin-2, gut-associated lymphoid tissue, lamina propria

## Abstract

Recent data indicate the link between the number and function of T regulatory cells (Treg) in the gut immune tissue and initiation and development of autoimmunity associated with type 1 diabetes (T1D). Since type 3 innate lymphoid cells (ILC3) in the small intestine are essential for maintaining FoxP3^+^ Treg and there are no data about the possible role of ILC3 in T1D pathogenesis, the aim of this study was to explore ILC3-Treg link during the development of T1D. Mature diabetic NOD mice had lower frequencies of IL-2-producing ILC3 and Treg in small intestine lamina propria (SILP) compared to prediabetic NOD mice. Similarly, in multiple low doses of streptozotocin (MLDS)-induced T1D in C57BL/6 mice, hyperglycemic mice exhibited lower numbers of ILC3, IL-2^+^ ILC3 and Treg in SILP compared to healthy controls. To boost T1D severity, mice were treated with broad-spectrum antibiotics (ABX) for 14 days prior to T1D induction by MLDS. The higher incidence of T1D in ABX-treated mice was associated with significantly lower frequencies of IL-2^+^ ILC3 and FoxP3^+^ Treg in SILP compared with mice without ABX treatment. The obtained findings show that the lower proportions of IL-2-expressing ILC3 and FoxP3^+^ Treg in SILP coincided with diabetes progression and severity.

## 1. Introduction

Type 1 diabetes (T1D) is an autoimmune disorder whose incidence has considerably increased over the past 20 years. This observation highlights a stronger influence of environmental factors on T1D development. Moreover, T1D is no longer considered as a childhood and adolescence disease, as it can also appear in the adult age [1]. A breakdown in self-tolerance to pancreatic β-cell antigens is a hallmark of T1D [2]. In T1D, β-cells are destroyed by the autoantigen-specific CD4^+^ and CD8^+^ T effector cells, leading to insulin deficiency [3]. T regulatory cells (Treg) maintain immune tolerance and a decrease or defective function in Treg has been demonstrated in T1D. Overall, T1D is characterized by the imbalance between effector T cells and the FoxP3^+^CD4^+^ Treg [4].

Apart from genetic predisposition, diet and changes in microbiota composition or intestinal permeability can influence T1D development [5,6]. Immune tolerance to food antigens and microbiota is maintained by the cells of gut-associated lymphoid tissue (GALT) [7]. In the circumstances of the impaired intestinal barrier, dietary ingredients such as β-casein or bovine insulin from cow’s milk, or even gluten, can break the oral tolerance in T1D [8]. Alterations in microbiota composition affect T1D incidence and development [9,10,11]. Individuals with T1D exhibit subclinical intestinal immune activation that may be an indirect proof of the relation between gut inflammation and pancreatic islets-directed autoimmunity [12]. Moreover, the activation of insulin-specific T cells can occur in the GALT and their presence has been confirmed in the Peyer’s patches and mesenteric lymph nodes almost at the same frequency as in the pancreatic lymph nodes [13].

A recent study has shown that the breakage of gut barrier continuity led to the activation of islet-reactive T cells in the intestine and the development of autoimmune diabetes. Therefore, the restoration of a healthy gut barrier through microbiota and diet modulation in diabetes-prone individuals could reduce intestinal activation of islet-reactive T cells and prevent T1D occurrence [14]. Miranda et al. [15] demonstrated that changes in the mucosal barrier precede the onset of disease in nonobese diabetic (NOD) mice and that Th17 cells and ILC3 were increased in the intestinal lamina propria, while the populations of tolerogenic dendritic cells (DC) and Treg were significantly reduced in the gut-draining lymph nodes which resulted in impairment of oral tolerance induction in NOD mice.

The role of innate lymphoid cells (ILC) in tolerogenic immune response in the gut in T1D is unknown. ILC are critical for the maintenance and regulation of mucosal homeostasis through the modulation of both innate and adaptive immunity [16,17]. The most recent data show the essential role of intestinal IL-2-producing ILC3 in the gut homeostasis and the induction of oral tolerance through the generation and maintenance of FoxP3^+^ Treg. The major population of IL-2-expressing cells in the small intestine lamina propria (SILP) is CD127^+^RORγt^+^ ILC3. The most prominent inducer of these cells is interleukin-1β (IL-1β), released from the intestinal macrophages in response to microbial cues [18].

The aim of this study was to examine the frequencies of IL-2-producing ILC3 and FoxP3^+^ Treg in SILP during the transition from prediabetes to diabetes in young and mature female NOD mice, during the development of disease in multiple low doses of streptozotocin (MLDS)-induced T1D, and to examine how treatment with broad-spectrum antibiotics before T1D induction affects these cell populations.

## 2. Results

### 2.1. SILP ILC3 and Treg in NOD Mice

Female NOD mice spontaneously develop T1D, becoming diabetic by 20 weeks of age. In order to establish whether the proportions of ILC3 and IL-2-expressing ILC3 change during the transition from prediabetes to diabetes, we assessed their frequencies along with Treg in the SILP in young (4 weeks of age) and mature (20 weeks of age) female NOD mice. ILC3 are CD45^+^ lineage-negative cells (Lin^neg^) that lack the lineage markers for T and B lymphocytes, monocytes/macrophages, granulocytes, NK cells and erythrocytes, and express RORγt and CD127. The obtained results showed that mature NOD mice had higher frequencies and absolute numbers of Lin^neg^CD45^+^RORγt^+^CD127^+^ ILC3 in the SILP compared to the young NOD mice (*p* = 0.006) (Figure 1A,B). However, the percentage of IL-2-expressing cells among Lin^neg^CD45^+^RORγt^+^CD127^+^ ILC3 was significantly lower in mature NOD mice compared to young NOD mice (*p* = 0.022) (Figure 1C,D), while serum IL-2 levels were similar in young and mature mice (Figure 1E). Further, mature NOD mice had significantly lower percentages and absolute numbers of CD4^+^CD25^hi^FoxP3^+^ Treg in the SILP compared to the young NOD mice (Figure 2A,B). Moreover, there was a significant alteration in the proportions of CD4^+^ and CD8^+^ T cells within the lamina propria, CD4^+^ cells being up-regulated, while CD8^+^ cells were down-regulated in older NOD mice (Figure 2C). Treg proportions in pancreatic lymph nodes (PLN) and pancreatic infiltrates did not differ between young and old NOD mice (Figure 2D), but the absolute numbers of Treg were lower in the PLN of older animals (Figure 2E).

The reduced proportions of IL-2-producing ILC3 and Treg were associated with the presence of insulitis in pancreatic islets of mature NOD mice (Figure 3B), while islets of young NOD mice were markedly less infiltrated by mononuclear cells (Figure 3A).

### 2.2. ILC3 and Treg in SILP of C57BL/6 Mice with MLDS-Induced Diabetes

To determine whether the same correlation between the presence of IL-2^+^ ILC3 and Treg within the SILP exists in another model of T1D, we chemically induced insulitis and subsequent hyperglycemia in C57BL/6 mice using MLDS. The obtained results in hyperglycemic C57BL/6 mice reveal lower proportion and absolute numbers of ILC3, IL-2^+^ ILC3 (Figure 4A,B) and lower numbers of Treg (Figure 5B), while the Treg frequencies were equal in comparison to normoglycemic healthy mice (control) (Figure 5A). The proportion of SILP CD4^+^ cells did not change during the diabetogenic process, but that of CD8^+^ T cells was significantly reduced (Figure 5C). Treg frequencies in the spleen and PLN were similar between the control and diabetic mice (Figure 5D), but their numbers were significantly lower in the diabetic group (Figure 5E).

### 2.3. The Effect of Antibiotics on SILP ILC3 and Treg in MLDS-Induced Diabetes

We next investigated how a change in gut microbiota prior to diabetes induction reflects on the presence of ILC3 and Treg in the SILP. Male C57BL/6 mice were treated with broad-spectrum antibiotics for 14 days, after which T1D was induced by MLDS. The significantly higher incidence of T1D was observed in antibiotics-treated C57BL/6 mice with MLDS-induced diabetes compared to the antibiotics non-treated animals (71% vs. 40%). Antibiotics-treated mice had higher glycemia compared to the non-treated animals, as shown in Figure 6. The higher T1D incidence in antibiotics-treated mice was associated with the significantly lower frequencies and absolute numbers of ILC3 and IL-2-expressing ILC3 in the SILP (Figure 7A,B). Furthermore, the frequencies and absolute numbers of FoxP3^+^ Treg in the SILP were markedly lower in antibiotics-treated animals compared to non-treated mice with MLDS-induced diabetes (Figure 7C,D), as well as the proportion of CD4^+^ T cells (Figure 7E).

We next investigated the serum concentration of IL-2 in C56BL/6 mice with MLDS-induced diabetes. We found that antibiotics-treated mice had a significantly lower concentration of IL-2 in the sera (*p* = 0.005) compared with the mice that were not treated with ABX prior to diabetes induction as shown in Figure 7F.

## 3. Discussion

Data obtained in this study show that the transition from prediabetes to diabetes in NOD mice is associated with the decrease in IL-2-producing ILC3 and FoxP3^+^ Treg in the SILP. Moreover, the development of hyperglycemia in C57BL/6 mice after streptozotocin administration coincides with a decrease in IL-2^+^ ILC3 and Treg numbers in SILP. This is the first report showing that the lower frequencies of IL-2-producing ILC3 in the small intestine are linked to T1D development. Our data indicate that IL-2^+^ ILC3 cells in the gut may play an important role in the regulation/prevention of T1D, the finding that warrants further investigation. Furthermore, changes in microbiota induced by the antibiotics treatment resulted in the higher incidence of MLDS-induced diabetes in C57BL/6 mice, which was associated with the reduction in IL-2-expressing ILC3 and Treg proportions in the SILP.

The incidence of autoimmune disorders has been dramatically rising worldwide. T1D is considered a polygenic disease with more than 40 loci linked to the disease risk in humans, including those coding for human leukocyte antigens [5]. The theory that genetic background is a major factor in the onset of autoimmunity is no longer sufficient to explain the pathogenesis of autoimmune diseases, and recent studies indicate that environmental factors may be triggers for T1D [6].

In T1D, the activation of autoreactive T cells results in the destruction of insulin-producing pancreatic β-cells, leading to insulin insufficiency. Studies in experimental models have shown that the environment contributes to T1D, including diet, exposure to infectious agents, and changes in microbiota composition, particularly in the gut. The study by Miranda et al. [15] has shown the importance of gut barrier integrity in the pathogenesis of T1D. The intestinal mucosa has the capacity to maintain immune tolerance to the self- and microbial-derived antigens while ensuring immunity against infection. Tolerogenic DC and T cells in the intestinal mucosa and lymph nodes in the gut promote oral tolerance, mediated by Treg, and the balance between tolerance and immunity is necessary for the gut homeostasis. Moreover, the authors showed that the changes in the mucosal barrier, including a reduction in the number of Goblet cells and impairment in MUC2 production, precede the onset of diabetes in NOD mice. They also found the local reduction in Th2 cells and type 2 innate lymphoid cells (ILC2) in the gut. Inflammatory cell subsets, such as Th17 cells and ILC3, were elevated in the lamina propria, while the populations of tolerogenic DC and Treg were significantly reduced in the gut-draining lymph nodes along with impaired oral tolerance in NOD mice. Our findings regarding the reduction in Treg in NOD mice are in agreement with this study. Moreover, our data suggest that older NOD mice (that exhibit severe insulitis) have altered frequencies of CD4^+^ and CD8^+^ cells in comparison to insulitis-free animals, suggesting that the diabetogenic process affects the gut immune system. Similar findings were observed in hyperglycemic C57BL/6 mice.

The importance of peripheral Treg in the control of T1D is supported by a number of animal studies. For example, peripherally-induced Treg have a crucial role in controlling the disease in NOD mice [19]. Moreover, it was shown that Treg isolated from young NOD mice have lower suppressive capacity compared to Treg from healthy Balb/c and C57BL/6 mice [20]. In pediatric T1D patients, blood PD-1^+^ Treg numbers were lower in comparison to healthy children [21]. The significance of intestinal regulatory cells in the modulation of pancreas inflammation was demonstrated by the finding that activated intestinal Tr1 regulatory cells migrate to the pancreas and suppress diabetogenic T cells in NOD mice [22]. Moreover, microbial products (such as short-chain fatty acids, SCFA) promote differentiation and activation of intestinal FoxP3^+^ Treg [23,24] and SCFA butyrate and acetate protect against T1D [25]. In line with these data, the absence of microbiota (caused by antibiotics) provokes accelerated insulitis with reduced FoxP3^+^ Treg in NOD mice [26], and in particular Treg within the intestinal tissue [27]. These results are in concordance with our findings that antibiotics aggravated MLDS-induced T1D in C57BL/6 mice and reduced the Treg numbers in SILP. As already defined in the literature [28], the number of Treg was elevated in the pancreas of older NOD mice suggesting that Treg infiltrate the inflamed tissue to combat against the T cell-mediated destruction. Having in mind the reduced numbers of Treg in other lymphoid organs (spleen and PLN) in older NOD and hyperglycemic C57BL/6 mice (our data and [29]), and the malfunction of Treg in NOD mice, it is reasonable to assume that Treg transplantation might have a beneficial effect for the mouse and human disease. To date, several studies have shown that Treg transplantation is effective in preventing or reversing the disease in mice [4]. These results prompted several phase I clinical trials where the transplantation of Treg or autoantigen-specific Treg was evaluated for their safety, as reviewed in [4].

Livanos et al. [27] have shown that therapeutic antibiotic pulses, based on the doses (and pharmacokinetics) used in young children, can accelerate T1D development and worsen insulitis when initiated early in life in NOD mice. This and other recent studies [30,31,32] highlight the importance of widespread antibiotic use, particularly in early childhood. However, there are studies showing the opposite effects of antibiotic use in T1D models [33,34], thus, antibiotic type, dose and timing can influence its effects in T1D. The same combination of antibiotics used in this study (ampicillin, vancomycin, neomycin sulfate and metronidazole) was shown effective in changing gut microbiota content in C57BL/6 mice. For example, they increased *Akkermansia*, *Parabacteroides* and *Verrucomicrobiales*, while decreased the proportion of *Bacteroides*, *Lactobacillus* and *Bifidobacterium*, as determined by 16S RNA sequencing [34]. Our data from the diabetic mice pretreated with antibiotics are consistent with the studies showing the key role of IL-2 in T1D [35]. Lower serum IL-2 in mice with a higher incidence of MLDS-induced diabetes when treated with antibiotics in our study are in line with the study that demonstrated marked decrease in serum IL-2 levels in patients with T1D at the time of diagnosis and in long-standing T1D [36]. 

The data regarding the role of ILC3 in T1D are scarce [37]. The protective effects of IL-2 involve the generation, maintenance and function of Treg. Until recently, the cellular and molecular pathways that control the production of IL-2 in the context of intestinal homeostasis were unknown. Zhou et al. [18] showed that IL-2 is acutely required to maintain Treg and immunologic homeostasis in the gastrointestinal tract. The authors revealed that ILC3 are the dominant cellular source of IL-2 in the small intestine, which is selectively induced by IL-1β. Intestinal macrophages produce IL-1β as a reaction to the microbiota. Further, they showed that ILC3-derived IL-2 is essential for the maintenance of Treg, immune homeostasis and oral tolerance to dietary antigens uniquely in the small intestine. In line with these findings, our results have shown that the development of T1D, either through a transition from prediabetes in NOD mice or after its induction by MLDS, is associated with a decrease in the frequencies of IL-2 producing ILC3. Moreover, the incidence of IL-2-producing ILC3 was reduced in C57BL/6 mice whose microbiota was changed by using antibiotics. 

The accumulating data point to the importance of gut barrier function and the onset of T1D. However, more knowledge needs to be gained about the gut–pancreas axis and its link to the autoimmune diabetes. The findings obtained in this study show that the decrease in IL-2-producing ILC3 and FoxP3^+^ Treg in the SILP is associated with the progression and higher incidence of T1D in mice. We anticipate that boosting of intestinal IL-2-expressing ILC3 in the presence of an autoantigen might provoke Treg-mediated oral tolerance and result in T1D prevention or amelioration. 

## 4. Materials and Methods

### 4.1. Mice

NOD/Ltj mice were purchased from Charles River (Italy), bred and maintained in a Uniprotect Air Flow Cabinet (ZOONLAB GmbH, Castrop-Rauxel, Germany). Male C57BL/6 mice (2–3 months old) were used for T1D induction as a susceptible strain [38,39]. Mice were bred and maintained at the Animal Facility at the Institute for Biological Research “Siniša Stanković”, University of Belgrade, with access to food and water *ad libitum*. All experiments were approved by the Ethical Committee of the Institute for Biological Research “Siniša Stanković” (App. No. 02-12/15) in accordance with the Directive 2010/63/EU.

### 4.2. T1D Induction and Treatment

T1D was induced in C57BL/6 mice by intraperitoneal injections of multiple low doses of streptozotocin (MLDS) for 5 consecutive days. Just prior to administration, streptozotocin (40 mg/kg bw, Sigma-Aldrich, St. Louis, MO, USA) was dissolved in cold 0.1 M citrate buffer (pH 6) [40]. Blood glucose levels were measured using a glucometer (Sensimac, IMACO GmbH, Lüdersdorf, Germany). Non-fasting blood glucose level above 11 mmol/L was considered as hyperglycemia. The incidence of diabetic animals was determined by dividing the proportion of hyperglycemic mice by the total number of mice. *Ex vivo* analyses were performed at day 20 after the first streptozotocin injection (for the comparison of healthy and hyperglycemic animals).

### 4.3. Antibiotic Treatment

Male C57BL/6 mice were treated with broad-spectrum antibiotics (ampicillin, vancomycin and neomycine sulfate, metronidazole) at 5 mg/day (5 mL per mouse of 1 mg/mL) for 14 days, after which T1D was induced by MLDS. *Ex vivo* cell analyses were performed on day 10 after the first streptozotocin injection.

### 4.4. Cell Isolation from SILP

Lamina propria mononuclear cells were isolated from the small intestine of young (4-week-old) or mature (20-week-old) female NOD mice, healthy C57BL/6 mice and antibiotics-treated or non-treated C57BL/6 mice (8–10 weeks old) with MLDS-induced T1D according to the amended protocol [41]. Briefly, mice were euthanized; the small intestine was removed and cut into pieces (approx. 5 cm long). After removing the intestinal content and the Peyer’s patches, the pieces were opened longitudinally, additionally cut into pieces (approx. 1 cm long) and washed thoroughly three times in cold phosphate buffered saline (PBS). Following this, the samples were washed in PBS containing 2% fetal calf serum (FCS) and 2.5 mM Dithiothreitol (DTT, Sigma-Aldrich) in an orbital shaker (250 rpm) for 15 min to reduce the presence of mucus. Next, the samples were washed three times in PBS containing 2% FCS and 5 mM EDTA (250 rpm, 15 min) to remove the intestinal epithelium cells. The tissue pieces were finally collected, washed in RPMI medium supplemented with 10% FCS (both from PAA Laboratories, Pasching, Austria) (250 rpm, 10 min) and resuspended in RPMI 10% FCS with Collagenase D (700 µg/mL) and DNase I (0.1 mg/mL) (both from Roche Diagnostics GmbH, Mannheim, Germany). The samples were incubated for 1 h at 37 °C in an orbital shaker (350 rpm). After digestion, the tissue was homogenized using a 70 µm cell strainer and centrifuged at 550× *g* for 5 min. The pellet was resuspended in a Percoll gradient solution (40%) and then layered upon 80% Percoll, after which it was centrifuged at 2000 rpm for 20 min without rotor brakes. SILP lymphocytes (found at the interface between 40% and 80% Percoll) were collected, washed twice in PBS and resuspended in RPMI 1640 supplemented with 5% FCS, 1% penicillin and streptomycin (all from PAA Laboratories, Pasching, Austria), 2 mM L-glutamine and 25 mM HEPES for further analysis.

### 4.5. Cell Isolation from Spleen and PLN

Spleen and pancreatic lymph nodes (PLN) were removed aseptically, and the cells were obtained by passing the tissue through a cell strainer (40 μm). If required, erythrocytes were first lysed with RBC lysis buffer (ThermoFisher Scientific, Waltham, MA, USA), after which the cells were resuspended in RPMI 5% FCS.

### 4.6. Cell Isolation from the Pancreatic Infiltrates

After removing the PLN, the pancreas was placed in Hank’s balanced salt solution (HBSS) (Sigma Aldrich). The residual fat was cleaned and the pancreas was cut into small pieces. The digestion of pancreatic tissue was carried out in collagenase type V (2 mg/mL, Sigma Aldrich) in HBSS containing 10% FCS, with shaking for 15 min at 37 °C. Digests were vortexed for 20 s and overlaid on density gradient media (HistoPaque-1083). After centrifuging at 600× *g* for 10 min, the cells between the layers were collected, washed twice and resuspended in RPMI 5% FCS.

### 4.7. Flow cytometric Analysis

For detecting Treg, surface staining was performed in PBS 1% BSA by adding the following anti-mouse antibodies: anti-CD4-FITC (rat IgG2b, κ), anti-CD4-eFluor 506 (rat IgG2a, κ) or anti-CD4-eFluor 450 (rat IgG2b, κ), and anti-CD25-PE (rat IgG1, λ) or anti-CD25-PeCy5.5 (rat IgG1, λ), as well as anti-CD8-APC (rat IgG2a, κ) (all from ThermoFisher Scientific). After 40 min at 4 °C, cells were washed twice and intracellular staining was performed using anti-mouse anti-FoxP3-PeCy5 (rat IgG2b, κ) or anti-Foxp3-PE (rat IgG2a κ) antibody (eBioscience, San Diego, CA, USA) and buffers from the Mouse Regulatory T cell Staining Kit (eBioscience) according to the manufacturer’s instructions. Isotype-matched controls were included in all experiments (eBioscience). Cells were analyzed on Partec CyFlow Space by FlowMax software (Partec, Görlitz, Germany). Cells were first gated on live cells (empirically determined) and further gated appropriately for the required analysis (Appendix A).

For detecting ILC3 and IL-2-producing ILC3, SILP cells were stimulated for 4 h with cell stimulation cocktail (that includes phorbol 12-myristate 13-acetate, ionomycin and brefeldin A) (ThermoFisher Scientific). The following anti-mouse antibodies for surface staining were used: mouse hematopoietic lineage antibody cocktail (recognizing CD3ε, CD11b, CD45R/B220, Ly-76/Ter119 and Ly-6G/Ly-6C)-PerCP-Cyanine5.5 (BD Biosciences, Bedford, MA, USA) or biotin-conjugated mouse lineage antibody cocktail (recognizing CD3ε, CD11b, CD45R/B220, Ly-76/Ter119 and Ly-6G/Ly-6C) coupled with streptavidin-APC-Cy7 (Biolegend, San Diego, CA, USA), anti-CD127-APC-eFluor 780 (rat IgG2a, κ), and anti-CD45-APC (rat IgG2b, κ) or anti-CD45-eFluor 506 (rat IgG2b, κ) (ThermoFisher Scientific). Following the FoxP3 protocol as described in the previous section, the following intracellular staining was performed using anti-mouse anti-RORγt-PE (rat IgG2a, κ) or anti-RORγt-PerCP-eFluor 710 (rat IgG2a, κ) and anti-IL-2-FITC (rat IgG2a, κ) or anti-IL-2-eFluor 450 (rat IgG2b, κ) antibodies (ThermoFisher Scientific). The gating strategy was as follows: viable cells were first gated on CD45^+^ and lineage-negative cells (Lin^neg^), then on CD127^+^ and finally on RORγt^+^ cells for total ILC3, and RORγt^+^ and IL-2^+^ for IL-2-producing ILC3 (Appendix A). Cells were analyzed by FACS Aria III (BD Biosciences) and FlowJo software.

### 4.8. ELISA

Serum samples were obtained from the blood by centrifugation at 12,000× *g* for 10 min. The concentration of IL-2 was determined by sandwich ELISA using MaxiSorp plates (Nunck, Rochild, Denmark) and anti-mouse paired antibodies (eBioscience) according to the manufacturer’s instructions. Samples were analyzed in duplicate and absorbance was measured by LKB microplate reader (LKB Instruments, Vienna, Austria) at 450 and 670 nm. A standard curve created from the known concentrations of recombinant IL-2 was used to calculate the concentration values in the samples.

### 4.9. Histology

Mouse pancreata were placed in 4% neutral buffered formalin, embedded in paraffin and processed by the routine histology procedures. For insulitis analysis, tissue sections were stained with Meyer’s hematoxylin (Bio-Optica, Milano, Italy).

### 4.10. Statistical Analysis

Data are presented as mean ± SD. A two tailed Student’s *t* test was used for determination of significance of differences between experimental groups, or a Mann–Whitney non-parametric test for the analysis of the data that were not normally distributed. If *p* < 0.05, the differences were considered as statistically significant. Statistical analyses were performed using GraphPad Prism 8 software (GraphPad Software, Inc., La Jolla, CA, USA).

## 5. Conclusions

The inflammatory process during T1D seems to correlate with the changes in the GALT immune cells, specifically the reduction of IL−2-producing ILC3 and Treg. These changes may be directly related to the autoimmune damage of the pancreas what requires further investigation.

## Figures and Tables

**Figure 1 molecules-28-03366-f001:**
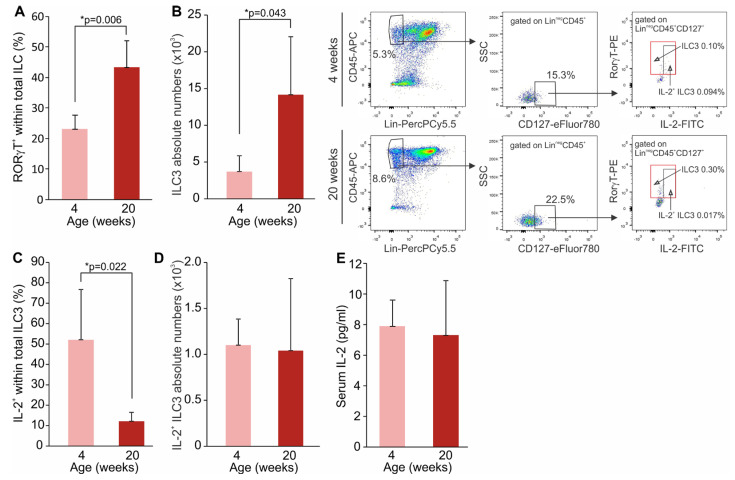
Mature NOD mice have lower frequencies of IL−2-producing ILC3 in the SILP. Proportion of total ILC3 (**A**) and their absolute numbers (**B**). The percentages of IL−2-producing cells among Lin^neg^CD45^+^CD127^+^RORγt^+^ ILC3 (**C**) and their absolute numbers (**D**) in the SILP of prediabetic (4-weeks-old) and diabetic (20-weeks-old) NOD mice. Representative dot plots are shown. Serum IL-2 concentration (**E**). Number of mice per group was 6. * *p* < 0.05 was considered as statistically significant when samples from mature and young mice were compared.

**Figure 2 molecules-28-03366-f002:**
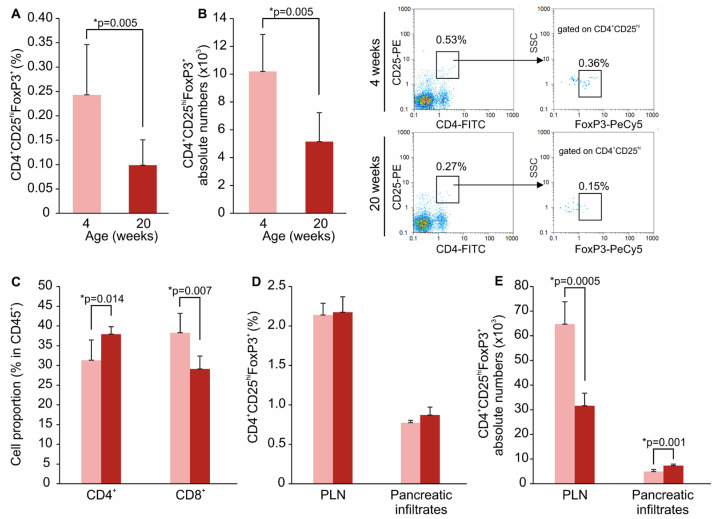
Mature NOD mice exhibit lower Treg proportions in the SILP. The percentages of CD4^+^CD25^hi^FoxP3^+^ Treg (**A**) and their absolute numbers (**B**) in the SILP of prediabetic (4-weeks-old) and diabetic (20-weeks-old) NOD mice. Representative dot plots are shown. Frequencies of CD4^+^ and CD8^+^ T cells in the SILP (**C**). Treg proportions (**D**) and absolute numbers (**E**) in the pancreatic lymph nodes (PLN) and pancreatic infiltrates. Number of mice per group was 6. * *p* < 0.05 was considered as statistically significant when samples from mature and young mice were compared.

**Figure 3 molecules-28-03366-f003:**
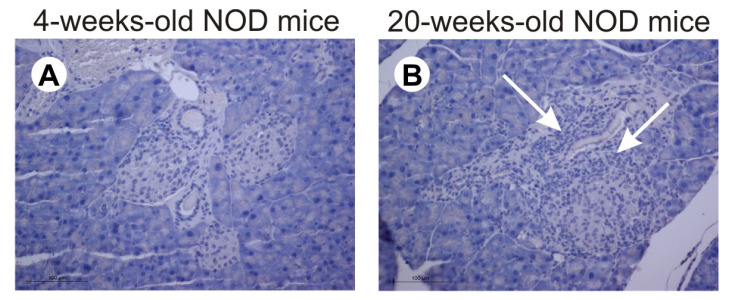
Insulitis is present in mature NOD mice (20-weeks-old). Representative images of pancreatic islets in prediabetic (**A**) and diabetic (**B**) NOD mice (arrows indicate the presence of immune cell infiltrates). Number of mice per group was 6. Number of inspected islets per section was 5.

**Figure 4 molecules-28-03366-f004:**
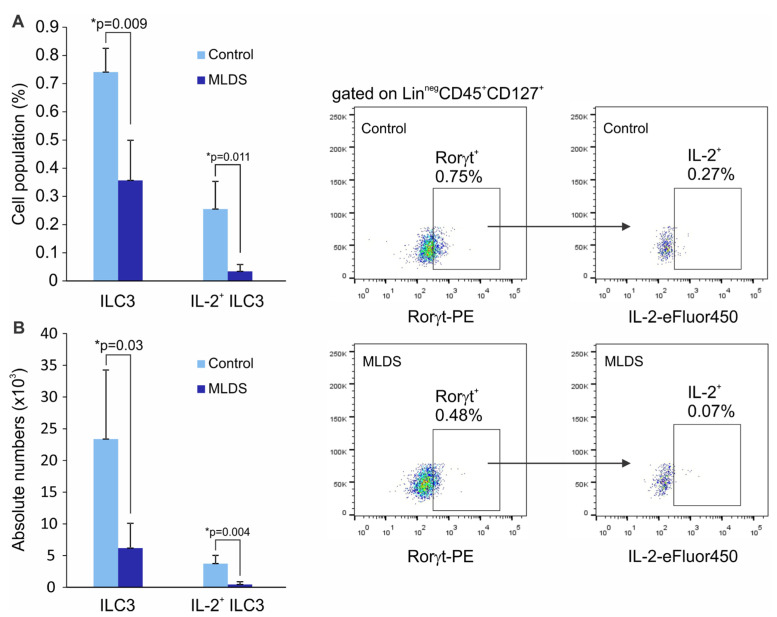
Diabetic C57BL/6 mice have lower frequencies of IL-2-producing ILC3 in the SILP. Proportion of total ILC3 and IL-2-producing cells among Lin^neg^CD45^+^CD127^+^RORγt^+^ ILC3 (**A**) in the SILP of normoglycemic healthy mice (control) and hyperglycemic C57BL/6 mice (MLDS), and their absolute numbers (**B**). Representative dot plots are shown. Number of mice per group was 5. * *p* < 0.05 was considered as statistically significant when samples from healthy and hyperglycemic mice were compared.

**Figure 5 molecules-28-03366-f005:**
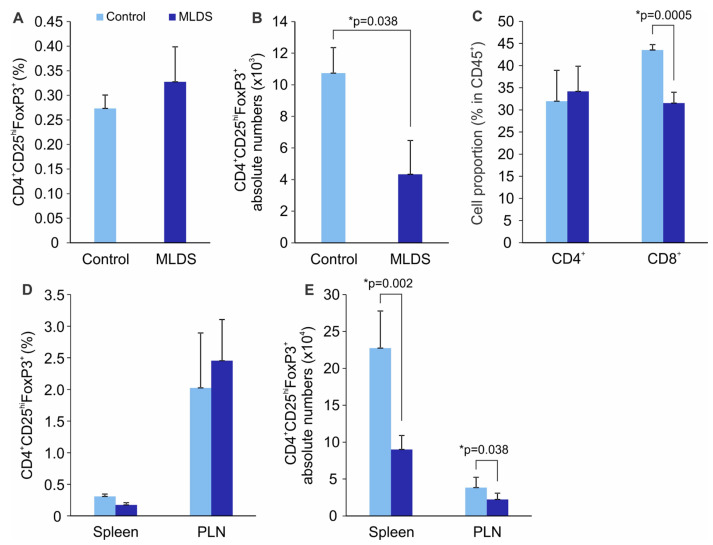
Diabetic C57BL/6 mice have lower numbers of Treg in the SILP and spleen. The percentages of CD4^+^CD25^hi^FoxP3^+^ Treg (**A**) and their absolute numbers (**B**) in the SILP of normoglycemic healthy mice (control) and hyperglycemic C57BL/6 mice (MLDS). Frequencies of CD4^+^ and CD8^+^ T cells in the SILP (**C**). Treg proportions in the spleen and pancreatic lymph nodes (PLN) (**D**) and their absolute numbers (**E**). Number of mice per group was 5. * *p* < 0.05 was considered as statistically significant when samples from healthy and hyperglycemic mice were compared.

**Figure 6 molecules-28-03366-f006:**
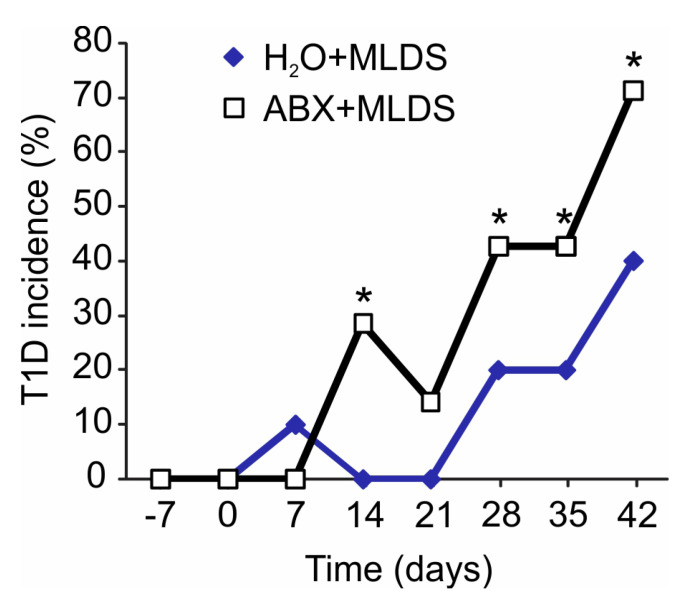
Incidence of T1D in antibiotics (ABX)-treated and non-treated C57BL/6 mice with MLDS-induced diabetes. Incidence (%) was calculated by dividing the number of mice with glycemia above 11 mmol/L with the total number of mice in the specific group (multiplied with 100). Day 0 on x axes represents the start of MLDS treatment. Number of mice per group was 10. * *p* < 0.05 was considered as statistically significant when samples from ABX+MLDS-treated and H_2_O+MLDS-treated mice were compared.

**Figure 7 molecules-28-03366-f007:**
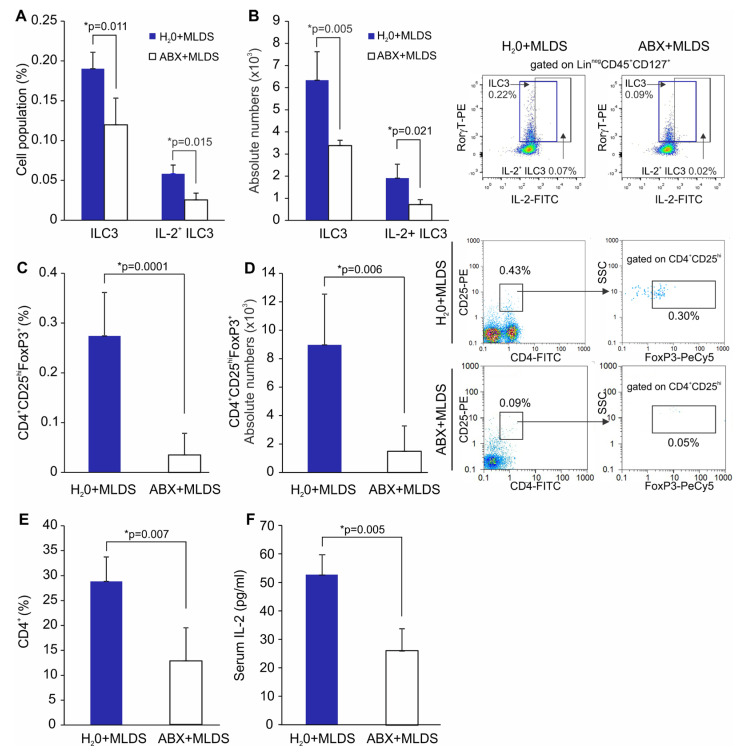
Accelerated MLDS-induced diabetes in antibiotics (ABX)-treated mice coincides with a decrease in IL−2-producing ILC3 and Treg in the SILP of C57BL/6 mice. The percentages and absolute numbers of Lin^neg^CD45^+^CD127^+^RORγt^+^ ILC3 and IL−2-producing ILC3 (**A**,**B**), CD4^+^CD25^hi^FoxP3^+^ Treg (**C**,**D**) and CD4^+^ (**E**) in the SILP of MLDS-induced diabetic C57BL/6 mice orally treated with ABX or H_2_O. IL−2 concentration in the sera of MLDS-induced diabetic C57BL/6 mice orally treated with ABX or H_2_O (**F**). Number of mice per group was 5. * *p* < 0.05 was considered as statistically significant when samples from ABX+MLDS-treated and H_2_O+MLDS-treated mice were compared.

## Data Availability

The data presented in this study are available on request from the corresponding author. The data are not publicly available due to privacy.

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
