# Peer review of "Development of Type 1 Diabetes in Mice Is Associated with a Decrease in IL-2-Producing ILC3 and FoxP3+ Treg in the Small Intestine"

_molecules, 2023, doi:10.3390/molecules28083366_

Round 1

Reviewer 1 Report

The manuscript entitled “Development of type 1 Diabetes in Mice is associated with a Decrease of IL-2-producing ILC3 and FoxP3+ Treg in the Small Intestine“ by Saksida et al. Is an excellent investigation into the role type 3 innate lymphoid cells play in regulating T regulatory cells in the gut ant its impact on the development of type 1 diabetes. Although the finding of the group is very compelling, a few additional items would really solidify the findings in the paper.
- Given the low number of events (which is to be expected for the tissue), gating can be a bit tricky. If clear positive events could be shown it would add a lot of weight to the gating. Activated CD4+ are a good source of IL2. Up to 10% of CD4s in the spleen are T-regs. Using these tissues to show a gating scheme for IL-2 and Foxp3+ cells would help solidify the gating.

- It may also be worth showing cell numbers. The principle reasoning for this is that Figure 5B. There is an unusual lack of CD4+ cells, so Tregs of a much reduced (potentially entirely reduced parental population), the raw number of event may be more informative.

-Lastly, given the excellent showing of two different models of T1D, it would be very impactful to show overlap of the two systems. For instance, evaluating the impact of antibiotics on NOD mice, or potentially using age matched controls of C57BL6 mice. The ages of the NOD are 4 and 20 weeks (very different stages of immunological development), where as the C57Bl6 were tested from 8-12 weeks. If the NOD mice could be treated with antibiotics, to evaluate if it accelerated the reduction of IL-2 in ILC3 cells it would nicely connect the two models.

Overall the manuscript identifies a very important aspect of gut immune cells and their role in T1D development. If the data can be shored up with the suggestions above, it would be hugely beneficial. Great work!

Author Response

- Given the low number of events (which is to be expected for the tissue), gating can be a bit tricky. If clear positive events could be shown it would add a lot of weight to the gating. Activated CD4+ are a good source of IL2. Up to 10% of CD4s in the spleen are T-regs. Using these tissues to show a gating scheme for IL-2 and Foxp3+ cells would help solidify the gating.

We thank the reviewer for the valuable suggestions.

We have included supplemental Figure S1 where we included plots with adequate isotype controls for Rorγt staining (Fig. S1A), for IL-2-eFluor450 staining in lamina propria cells (for the additional experiment that we performed) (Fig. S1B), for IL-2-FITC staining in spleen cells for the experiments in NOD and in ABX-treated C57BL/6 mice (as the reviewer suggested) (Fig. S1C), and for FoxP3 in spleen cells (Fig. S1D).

- It may also be worth showing cell numbers. The principle reasoning for this is that Figure 5B. There is an unusual lack of CD4+ cells, so Tregs of a much reduced (potentially entirely reduced parental population), the raw number of event may be more informative.

The total numbers of cells isolated from lamina propria was 2,4±0,8x106 vs 4,0±0,5x106 (MLDS vs ABX+MLDS). In the revised version of the manuscript, we have included absolute numbers for ILC3 and Treg analysis. In addition, we have substituted dot plot in Fig 5 with the more appropriate one (as we have shown the lowest number of CD4 and Treg, i.e. 5%). The average proportion of CD4+ cells is indeed significantly lower in ABX-treated mice (Fig. 7E). This reduction in CD4+ frequency is in accordance with the literature data (Ekmekciu et al., 2017).

-Lastly, given the excellent showing of two different models of T1D, it would be very impactful to show overlap of the two systems. For instance, evaluating the impact of antibiotics on NOD mice, or potentially using age matched controls of C57BL6 mice. The ages of the NOD are 4 and 20 weeks (very different stages of immunological development), where as the C57Bl6 were tested from 8-12 weeks. If the NOD mice could be treated with antibiotics, to evaluate if it accelerated the reduction of IL-2 in ILC3 cells it would nicely connect the two models.

This is an excellent idea to treat NOD mice with antibiotics, as literature data show that prolonged antibiotic treatment induces a diabetogenic intestinal microbiome that accelerates diabetes in NOD mice (Brown at el., 2016; Livanos et al., 2016). Also, there is data that germ-free environment accelerates and aggravates T1D progression in NOD mice (Alam et al., 2011). In the study by Livanos et al., 2016, they have stated that several administrations of antibiotics early in life of a NOD mouse (that mimic childhood exposures) leads to reduced Th17 and Treg proportions in the intestinal lamina propria. As this was already published (and indicated in Discussion section), we feel that the repetitions of such experiment would be redundant. However, to strengthen our study, we decided to introduce another test system, i.e. to compare ILC3 and Treg in healthy versus hyperglycemic C57BL/6 mice. Our data suggest that hyperglycemia correlates with down-regulation of ILC3 and Treg in the small intestine lamina propria and corroborates the data obtained in the old NOD mice with severe insulitis.

Overall the manuscript identifies a very important aspect of gut immune cells and their role in T1D development. If the data can be shored up with the suggestions above, it would be hugely beneficial. Great work!

Thank you!

Reviewer 2 Report

The authors in this study show that the induction of type-1 diabetes(T1D) in NOD mice corelates with the corresponding reduction in IL-2 producing ILC-3 and consequently reduced Tregs in the SI-LP. The proposed hypothesis is that the loss of Tregs would lead to activation of auto-reactive T cells  damaging the pancreatic insulin producing b-cells, thereby leading to T1D.

There are several points for improvement in this study:

1.     The entire study remains correlative There is no direct evidence to show that there is increased inflammation in the pancreas is indeed due to the loss of Tregs in SI. What are the frequencies of other T cell subsets like CD4, CD8, DC in this analysis.

2.     Also, do the frequency of Tregs change in other lymphoid organs like spleen and lymph nodes?

3.     The absolute number of cells needs to be show along with the percentages. Particularly, since there is significant difference in the frequencies of ILC3 in the 4weeks and 20-week-old NOD mice.

4.     The flow plots are not convincing. The staining probably has not worked effectively. I say this because the frequencies of ILCs and Tregs is much lower than what is generally observed in control mice. Infact, the IL-2 staining probably is not real. The lymphocytes need to be re-stimulated in presence of golgi block to effectively detect any cytokine.

5.     It is important to show that antibiotic treatment reduced the commensal burden in the mice.

6.     Would adoptively transfer of Tregs rescues the mice phenotype.

Author Response

1.     The entire study remains correlative There is no direct evidence to show that there is increased inflammation in the pancreas is indeed due to the loss of Tregs in SI. What are the frequencies of other T cell subsets like CD4, CD8, DC in this analysis.

Thank you for all the suggestions. We have implemented the response to your comments either in the Result or Discussion section of the revised manuscript.

The reviewer is right about the correlative nature of this study. In order to get deeper insight in the relationship of ILC3 and Treg in the small intestine lamina propria and the development of T1D, we are preparing a project proposal for adequate financial resources that will cover knockout mice for IL-2 specifically in NKp46+ cells (one of the possible markers of ILC3). In order to strengthen this current study, we performed another experiment where we compared healthy C57BL/6 mice with hyperglycemic MLDS-treated mice. The results show a similar trend as in other two test systems, i.e. IL-2+ ILC3 and Treg absolute numbers in the lamina propria were reduced in hyperglycemic animals compared to healthy ones.

In the revised version of the manuscript, we have included frequencies of CD4+ and CD8+ within the SILP. We have found an increase in CD4+ and decrease in CD8+ cell frequencies in SILP of older NOD mice. Also, there was an increase in CD8+ cells in hyperglycemic C57BL/6 mice. As for ABX-treated T1D mice, they exhibited significant decrease in CD4+ in SILP which is in accordance with the literature data (Ekmekciu et al., 2017).

2.     Also, do the frequency of Tregs change in other lymphoid organs like spleen and lymph nodes?
The numbers of Treg are decreased in the spleen and pancreatic lymph nodes in the test systems that we have investigated. These graphs are now included in the revised Figures. Also, we have included the numbers of Treg within the pancreatic infiltrates that were elevated in older NOD mice with insulitis. This is in accordance with the literature data (Kaur et al., 2010). We have implemented this discussion in the revised version of the manuscript.

3.     The absolute number of cells needs to be show along with the percentages. Particularly, since there is significant difference in the frequencies of ILC3 in the 4weeks and 20-week-old NOD mice.

The absolute numbers of ILC3 and Treg are included in the revised version of the manuscript in all Figures.

4.     The flow plots are not convincing. The staining probably has not worked effectively. I say this because the frequencies of ILCs and Tregs is much lower than what is generally observed in control mice. Infact, the IL-2 staining probably is not real. The lymphocytes need to be re-stimulated in presence of golgi block to effectively detect any cytokine.

Cells isolated from small intestine lamina propria were subjected to cell stimulation cocktail (that includes phorbol 12-myristate 13-acetate, ionomycin and brefeldin A), for 4h before staining. We apologize for omitting this information from the first version of the manuscript. In order to choose proper gating we were using the adequate isotype controls and we have included dot plots showing the gating strategy in the revised version of the manuscript (Figure S1).

5.     It is important to show that antibiotic treatment reduced the commensal burden in the mice.

Unfortunately, we did not perform 16S RNA sequencing, but literature data indicate that the same combination of antibiotics (ampicillin, vancomycin, neomycin sulfate and metronidazole), at the same concentration that we have applied, to the same strain of mice (C57BL/6 mice) was effective in changing gut microbiota content (for example, it increased the proportion of Akkermansia, Parabacteroides and Verrucomicrobiales, while decreased the proportion of Bacteroides, Lactobacillus and Bifidobacterium), as determined by 16S RNA sequencing (Costa et al., 2016).

6.     Would adoptively transfer of Tregs rescues the mice phenotype.

Having in mind the reduced numbers of Treg in other lymphoid organs (spleen and PLN) in older NOD and hyperglycemic C57BL/6 mice (our data and Nti et al., 2012), and the malfunction of Treg from NOD (Godoy et al., 2020), it is reasonable to assume that Treg transplantation might have a beneficial effect for the both the mouse and human disease. To date, several studies have been shown that Treg transplantation is effective in preventing or reversing the disease in mice (Yu et al., 2018). Also, several phase I clinical trials have been ensued where the transplantation of Treg or autoantigen-specific Treg was evaluated for their safety (Tang et al., 2004; Marek-Trzonkowska et al., 2012; Marek-Trzonkowska 2016). This information is introduced in the Discussion section of the revised manuscript.

Round 2

Reviewer 2 Report

All the comments have been addressed and  are acceptable.